# Design and Analysis of the Task Distribution Scheme of Express Center at the End of Modern Logistics

**Chunxue Wu [1], Junjie Wu [1], Yan Wu [2], Qunhui Wu [3], Xiao Lin [4] and Neal N. Xiong [5,*]**

[1] School of Optical-Electrical & Computer Engineering, University of Shanghai for Science and Technology, Shanghai 200093, China; wcx@usst.edu.cn (C.W.); monday94@126.com (J.W.)

[2] O'Neill School of Public and Environmental Affairs, Indiana University, Bloomington, IN 47405, USA; wuyan8910@gmail.com

[3] Shanghai Hao long environmental technology Co., Ltd., Shanghai 201610, China; Hest2016@126.com

[4] Department of Computer Science, Shanghai Normal University, Shanghai 200234, China; lin6008@126.com

[5] Department of Mathematics and Computer Science, Northeastern State University, IN 200093, USA; xiongnaixue@gmail.com

* Correspondence: xiongnaixue@gmail.com; Tel: +86-137-6703-9700

**Abstract:** With the rise and improvement of artificial intelligence technology, the express delivery industry has become more intelligent. At the terminal of modern logistics, each dispatch center has hundreds of express mail deliveries to be dispatched every day, and the number of dispatchers is far less than the number of express mail deliveries. How to assign scientific tasks to each courier dispatch is the main target of this paper. The purpose is to make the number of tasks between the various couriers in the express center roughly the same in each cycle, so that there is a more balanced income between the couriers. In the simulation experiment, the delivery addresses are clustered according to the balanced k-means algorithm. Then, the ant colony algorithm is used to plan the delivery order of the express items in each class. Then, the time cost model is established according to the delivery distance of the express items in each class and the delivery mode of the express items to calculate the delivery time cost. Through a large amount of experimental data, the standard deviation of delivery time cost of each courier gradually decreases and tends to stabilize, which suggests that this method has a good effect on the dispatching task assignment of the express center. It can effectively make the delivery workload between the distributors roughly the same, and improve the delivery efficiency of the courier, save energy, and promote sustainable development.

**Keywords:** express delivery center; task assignment; dispatch; time cost

## 1. Introduction

With the rise and development of online shopping, peoples' consumption has changed greatly. More and more people choose to conduct shopping online, which greatly promotes the development of the express industry. After investigation, the author learned that, before the express mail reached the delivery address, it would be transported to the regional express center, and then assigned to specific couriers for delivery. The express center needs to deliver far more express items than the number of couriers every day, and how to distribute these items to couriers to make the workload of each courier more balanced has become a problem. The size of the workload will directly affect the salary of each courier. If the workload of couriers is different, their salaries will be different. In addition, in reality, employees who work more often get less pay, while those who work less often get more pay, which is not conducive to the development of courier companies. Therefore, how to reasonably allocate the workload of couriers has become a very important and practical objective.

At present, the courier center assigns the courier that needs to be delivered on the same day to each courier. It is often distributed according to their experience, which leads to the imbalance of the assigned tasks. The imbalance of task allocation often affects the courier. There is a problem with the distribution of wages. The purpose of this paper is to study a set of task assignment methods for express delivery personnel to the courier center, in order to solve the problem that the courier center assigns unbalanced task assignments to the courier, so that the courier center can more appropriately distribute the work of each courier. Quantity and income, can not only improve the delivery efficiency of the express center, but also promote the development of the courier company.

The balanced k-means algorithm [1] is used to cluster [2] each delivery address, and then the ant colony algorithm is used to plan the delivery order of the shipments in each class, and then calculate the delivery time cost according to the delivery route of each class and the delivery mode of each shipment. On the first day, according to the time cost of each class, it is assigned to the courier from low to high, and the courier's accumulated time cost is sorted. On the following day, the class with the time cost from low to high is assigned to the courier who accumulates the cumulative cost of the time from high to low. Through a large amount of experimental data, it can be known that this method has a good effect on the assignment of dispatching tasks to the express center [3–5], which can solve the problem that the express center assigns unbalanced tasks to the courier, so that the express center can distribute more reasonably. The workload and income of each courier can not only improve the delivery efficiency of the courier center, but also promote the development of the courier company [6,7].

The rest of this paper is organized as follows. Section 2 describes the research progress related to this paper, explains the application of the ant colony algorithm and genetic algorithm in path planning, introduces the driving factors and future prospects of the express delivery industry, analyzes a variety of task assignment methods, and compares and introduces a variety of clustering algorithms. Section 3 first explains the principle of the k-means and balanced k-means algorithms, and their application in this paper, and then introduces the principle and steps of the ant colony algorithm. Finally, based on the research background of this paper, the time cost model of courier delivery, and its application principle in this paper, are established. Section 4 describes the performed simulation experiment. By using the balanced k-means algorithm, ant colony algorithm, and time cost model, the standard deviation of each courier's cumulative delivery time cost in a period of time is obtained, and the feasibility of the task assignment method constructed in this paper is demonstrated through analysis.

## 2. Related Works

Russell G. Thompson [8,9] considered the cost of driving and walking when studying the optimal delivery route, and proposed a two-layer optimization model, which simultaneously generates a driving dispatch route and a pedestrian delivery route, and proposes a simple and efficient method. The genetic algorithm is used to search for the optimal solution.

Tsung-Sheng Chang [10] proposed an innovative route scheduling strategy to help city express delivery operators to reduce operating costs and improve their service levels. This scheduling strategy considers urban route scheduling as a static problem by considering the specific characteristics of the urban street network, and specifically studies the fixed service location and time.

Raphaëlle Ducret [11–13] used France as a case to study the changes and challenges of the European express delivery industry, analyzed the driving factors of the express delivery industry, and highlighted the future prospects of the new segment of the express delivery industry.

A.V. Hill [14] studied a decision support system for express vehicle dispatching, which helps to correctly determine the number of vehicles, appropriate route collection, and route travel time. The system has been implemented in some companies, proving its practical application value.

Felipe Balmaceda [15,16] studied an optimal task assignment method with a loss avoidance agent. It shows that loss avoidance combines the marginal cost of work in one task with the increase in the workload selected in other tasks in order to determine when multitasking saves compensation costs.

Shima Rashid [17] studied mobile cloud task allocation based on a mixed heuristic queue, proposed a multi-server queue model for extracting the system performance parameters, and proposed a resource allocation mechanism for two-level mobile cloud computing architecture with an unloading mechanism.

Chyh-Ming Lai [18] used the entropy simplified group optimization algorithm to study the task assignment problem, which uses entropy to describe the uncertainty level of the assigned task, so that tasks with a higher uncertainty have more opportunities to be redistributed.

G. Melendez-Melendez [19] proposed an improved partial clustering algorithm, which shows that outlier detection is very important for improving clustering results. The excessive detection of outliers leads to information loss. Experiments have shown that clustering quality can be improved while reducing run time.

Ahsan Jalal [20] proposed a semi-supervised clustering method for unknown expressions, using deep CNN (convolutional neural network) to learn to embed facial behaviors that help cluster unknowns. In the experiment, it was found that facial aggregation captures not only facial expressions, but the intensity of some expressions too.

Geng Zhang [21] proposed an improved K-means algorithm based on the density canopy, which improves the accuracy and stability of the K-means algorithm and solves the problem of determining the most suitable cluster number K and the optimal initial seed.

Keivan Bamdad [22] proposed an ant colony optimization scheme for continuous domains when using the ant colony algorithm in order to establish energy optimization problems, and applied it to optimize commercial buildings in Australia. His research indicates that this optimization scheme can further promote the design of low-energy buildings.

Mohamed MS Abdulkader [23] used a hybrid ant colony algorithm to study multi-chamber vehicle routing problems. He evaluated the performance of the algorithm based on numerical experiments. Abdulkader combined the local search with the existing ant colony algorithm and compared it with the existing ant colony algorithm. The experimental results show that the mixed ant colony algorithm can improve the performance of the algorithm.

Yongbo Li [24] proposed a multi-objective linear mathematical model in order to solve the problem of target conflict by using the ant colony algorithm to study the multi-objective vehicle routing problem based on maximizing revenue and minimizing cost. The effectiveness of the model was verified by experiments.

Saeed Rubaiee [25] proposed an energy-aware multi-target ant colony algorithm to minimize the total completion time and energy cost of stand-alone preemptive scheduling. He considered electricity prices and scheduling goals in experimental scheduling and adopted sustainable thinking in order to balance energy needs and power generation capabilities.

Raka Jovanovic [26] proposed an ant colony optimization algorithm for the lifting block positioning problem, which is suitable for both unrestricted and different deadline restrictions. In the unrestricted case, Raka Jovanovic proposed a new heuristic method and defined it as a greedy algorithm.

Nikolaos D. Doulamis [27] proposed an algorithm for assigning tasks to resources that minimizes the violations of the tasks' time requirements while simultaneously maximizing the resources' utilization efficiency for a given number of resources. The results show that the scheduling performance of this algorithm is better than other algorithms under different granularity values and task request loads.

By referring to the above and other references, the author learned that the balanced k-means algorithm is easy to realize in clustering, and the clustering effect is outstanding. Therefore, this paper adopts the balanced k-means algorithm to cluster the express items that need to be sent according to the delivery address. The value of K is the number of couriers.

When the courier delivers the express mail, it is essential to traverse the address of the delivery. In essence, it can be regarded as a problem of travel agents. After reading the above reference documents, the author learned that the ant colony algorithm has strong robustness and a global search ability in solving the TSP (Traveling Salesman Problem) problem, so this paper adopted the ant colony algorithm to solve the problem of the delivery order of couriers.

## 3. Algorithm Introduction

### 3.1. k-means and Balanced k-means Clustering Algorithm Introduction

The k-means clustering algorithm [28–31] has a high efficiency in data clustering [32]. Therefore, it is used to cluster the daily deliveries of express delivery centers based on addresses.

The k-means algorithm is a very common clustering method [33,34]. When it is clustered, its time complexity is only O(tkn), where t is the number of iterations of the algorithm, k is the number of clusters required in the algorithm, and n is the total amount of data in the algorithm. In the simulation experiment mentioned later, t is the number of iterations of the algorithm operation until the optimal solution is calculated. The value of this experiment is 1000, k is the number of couriers who can send the goods on the same day, and n is the number of shipments that the delivery center needs to dispatch that day [35].

The k-means algorithm first randomly selects K objects as the initial cluster center. The distance between each object and each seed cluster center is then calculated, and each object is assigned to the cluster center closest to it. The cluster centers and the objects assigned to them represent a cluster. Once all objects have been assigned, the cluster center of each cluster is recalculated based on the existing objects in the cluster. This process will continue to repeat until a termination condition is met. The termination condition may be that no (or a minimum number of) objects are reassigned to different clusters, no (or a minimum number of) cluster centers are re-changed.

The goal of the k-means algorithm is to divide n data points into k clusters, find the center of each cluster, and let the points in the cluster gather together as closely as possible, so that the points between the clusters are as far away as possible. Assuming that the cluster is divided into $C_1, C_2, \ldots, C_k$, the goal of the algorithm is to minimize the squared error E, and its mathematical expression is:

$$E = \sum_{i=1}^{k} \sum_{C_i} \left\| x - \mu_i \right\|_2^2, \tag{1}$$

where $\mu_i$ is the mean vector of the cluster $C_i$, also known as the centroid of the cluster $C_i$, and its mathematical expression is:

$$\mu_i = \frac{1}{|C_i|} \sum_{x \in C_i} x, \tag{2}$$

According to the formula above, it can be known that the k-means algorithm is to solve an NP-hard (non-deterministic polynomial) problem. It is impossible to solve such a problem directly as they can only be solved by heuristic iterative methods.

When clustering coordinate points in a European plane, the most commonly used optimization criterion is the mean square error (MSE):

$$MSE = \sum_{j=1}^{k} \sum_{X_i \in Cj} \frac{\left\| X_i - C_j \right\|^2}{n}, \tag{3}$$

where $X_i$ is the data point position, and $C_j$ is the center of mass position.

The k-means algorithm has two basic operations: assignment and update.

Assignment: Place the data points being processed into the nearest centroid dependent cluster.

$$P_j^{(t)} = \left\{ X_i : \left\| X_i - C_j^{(t)} \right\| \leq \left\| X_i - C_{j^*}^{(t)} \right\| \forall j^* = 1, \ldots, k \right\}, \tag{4}$$

Update operation: Calculate and update the mean of each cluster.

$$C_j^{(t+1)} = \frac{1}{|P_j^{(t)}|} \sum_{X_i \in P_j^{(t)}} X_i, \tag{5}$$

The assignment operation of the balanced k-means [36] algorithm constructs a weight function based on the k-means algorithm, with the goal of minimizing the cost function [37].

$$\text{Cost} = \sum_{a \in A} W(a, f(a)). \tag{6}$$

The update operation of balanced k-means algorithm is similar to that of the k-means algorithm.

$$C_i^{(t+1)} = \frac{1}{n_i} \sum_{X_i \in C_i^{(t)}} X_j, \tag{7}$$

The weight construction formula of the edge is:

$$W(a, i) = \text{dist}\left(X_i, C_{(a \bmod k)+1}^t\right)^2 \forall a \in [1, n] \ \forall_i \in [1, n], \tag{8}$$

After the algorithm converges, the partition standard of point $X_i$ is:

$$X_{f(a)} \in P_{(a \bmod k)+1}, \tag{9}$$

## 3.2. Ant Colony Algorithm for TSP Problem

The ant colony algorithm [38–43] is used to solve the order of shipments in each cluster and to dispatch the shortest path.

The steps of the ant colony algorithm in solving the optimal path are as follows:

Step 1: Initialize relevant parameters, including ant colony size, pheromone factor, heuristic function factor, pheromone volatile factor, pheromone constant number, maximum iteration number, etc., and read the data into the program for preprocessing.

Step 2: Randomly place the ants at different starting points and calculate the next visiting city for each ant until there are ants accessing all the cities.

Step 3: Calculate the path length $L_k$ of each ant, record the optimal solution of the current iteration number, and update the pheromone concentration on the path.

Step 4: Determine whether the maximum number of iterations is reached. If not, return to Step 2—otherwise end the procedure.

Step 5: Output the results, and output relevant indicators in the optimization process, such as running time, convergence iterations, and the shortest distance, as needed.

In solving the traveling salesman problem [44,45] (TSP), the ant colony algorithm should give the distance between two different cities, and finally find the shortest path through each city.

The first step is initialization.

M ants were randomly placed in n cities. The taboo of each ant was the city in which the ant was currently located, and each piece of edge information was initialized to c. The taboo table reflects the memory of artificial ants, so that ants will not take a repetitive path and improve efficiency.

The second step: construct the path.

At time t, the probability that ant $k$ moves from city $i$ to city $j$ is:

$$p^k(i,j) = \begin{cases} \frac{[\tau(i,j)]^\alpha \cdot [\varphi(i,j)]^\beta}{\sum_{s \in J_k(i)} [\tau(i,s)]^\alpha \cdot [\varphi(i,s)]^\beta}, & \text{if } j \in J_k \\ 0 & \text{else} \end{cases}, \tag{10}$$

where Tab $u_k$ is the collection of cities that each ant $k$ has visited, $J_k = \{N - \text{Tab}u_k\}$; $\alpha$, $\beta$ is the system parameter, respectively representing the influence degree of pheromone and distance on ants' path selection; $\tau(i, j)$ represents the concentration of pheromones on edge $L(i, j)$, and $\varphi(i, j)$ indicates that, depending on the expected degree of city $i$ to city $j$, the class can be specifically determined according to the heuristic algorithm, generally $\frac{1}{d_{ij}}$.

The larger the information heuristic factor $\alpha$ is, the greater the probability that ants will choose the path they have traveled before, and the less random the search path is. The smaller $\alpha$ is, the smaller the search range of the ant colony will be, making the algorithm prone to fall into local optimization. When A = 0, the algorithm evolves into the traditional random greedy algorithm, and the nearest city has the highest probability of being selected. The larger the expected heuristic factor is, the easier it is for the ant colony algorithm to choose the local shorter path. At this time, the convergence speed of the algorithm is accelerated, but the randomness is weakened, and it is easy to get the local relative optimal. When Beta = 0, the ant completely determines the path according to the pheromone concentration, and the algorithm will converge rapidly, so that the constructed path is far from the actual target. A large number of experiments have shown that it is more appropriate to set $\alpha$ = 1~2 and beta = 2~5.

The third step: update pheromones.

After all ants find a legitimate path, the pheromone is updated (ant week model).

$$\tau_{ij}(t+1) = (1 - \rho)\tau_{ij}(t) + \sum_m \Delta\tau_{ij}^k(t, t+1), \tag{11}$$

$$\Delta\tau_{ij}^k(t, t+1) = \begin{cases} \frac{Q}{L_k}, & \text{pass } (i, j) \\ 0, & \text{not pass } (i, j) \end{cases}, \tag{12}$$

where $\rho$ is the volatile factor of the pheromone, and its value is less than 1 positive number, generally 0.5. So, on the one hand, it is to prevent the infinite accumulation of pheromones and, on the other hand it is to improve the system search for a more feasible solution ability, in order to avoid the earlier lost ability to explore new paths. Because the volatile factors of pheromones are too small, there are too many pheromones left on each path, which leads to the search of invalid paths, and affects the convergence speed of the algorithm. However, if the volatile factor of pheromones is too large, invalid paths can be excluded from searching, but there is no guarantee that effective paths will not be abandoned from searching, which will affect the search for the optimal value. Further, m represents the number of ant colonies. The greater the value of m, the more accurate the optimal solution will be, but many repeated solutions will be generated. As the algorithm approaches the optimal value of convergence, the positive feedback effect of information will be reduced, and a large number of repeated works will consume resources and increase the time complexity.

$\Delta\tau_{ij}$ represents the pheromone intensity left by ants in path $L(i, j)$ during this operation.

$\Delta\tau_{ij}^k$ represents the pheromone intensity of ant $k$ placed on edge $L(i, j)$.

Q represents the normal number of tracks left by ants (10, 10,000).

$L_k$ represents the total length of the path traveled by the $k$ ant in this tour.

The fourth step: output results.

If the number of iterations is less than the predetermined number of iterations and there is no degradation behavior (all found are the same solution), then go to Step 2—otherwise the current optimal solution will be output.

The ant colony algorithm has three updating models in the third pheromone update: The ant cycle model, ant quantity model and ant density model.

The differences between the three pheromone renewal models are as follows:

Ant-period model [46]: The pheromone increment is $Q/L_k$, which is only related to the search route and has nothing to do with the specific path $(i, j)$. After the KTH ant completes a path search, the pheromone updates all the paths on the route. The pheromone increment is related to the overall line of this search, so it belongs to the global information update.

Ant quantity model [47]: The pheromone increment is $Q/d_{ij}$, which is related to the length of the specific path $(i, j)$. During the progress of the ant colony, the ant updates the pheromone on the path after completing one step of movement. Use ants to walk through the information on the path $(i, j)$ for updates belonging to local information updates.

Ant density model [48]: The pheromone increment is a fixed value Q. During the progress of the ant colony, the ant updates the pheromone on the path after completing one step of movement. Use ants to walk through the information on the path $(i, j)$ for updates, belonging to local information updates.

### 3.3. Our Proposed Delivery Time Cost Algorithm

This paper presents an algorithm for calculating the delivery time cost of express delivery, referred to as the delivery time cost algorithm.

Before the task assignment, the express center first clusters the shipments that need to be delivered on the same day according to the address, and then calculates the total delivery time cost of each class, and finally assigns the shipments in each class to the courier according to the time cost [49–51].

When establishing the express delivery time cost model, it mainly depends on the delivery route of each class and the receiving method of each shipment address. The delivery distance is the shortest delivery distance that the courier gets from the courier center and needs to send the courier, and then delivers the courier according to the optimal delivery order. The delivery method of each shipment mainly considers two types that are commonly used nowadays, namely delivery to home (DTH) and the courier deposit to the nearby courier to be accepted by the consignee (EC).

Step 1: Calculate the total distance of delivery, then divide the distance of delivery by the average speed of delivery, and get the time needed by the delivery personnel during the delivery.

Step 2: Calculate the number of express deliveries needed to be delivered and multiply it by the average delivery time.

Step 3: Count the number of deliveries to the express cabinet and multiply it by the average delivery time.

Step 4: Add all the above time costs to get the total cost of delivery time.

The time cost model established is:

$$\text{T} = \frac{\sum d_{ij}}{v} + \text{n}T_s + mT_t \, , \tag{13}$$

In the above formula, T represents the total cost of delivery time for each express class, $d_{ij}$ represents the distance between the adjacent delivery addresses that the dispatcher has traveled, $v$ represents the average speed of the delivery of the courier, and $\text{n}T_s$ indicates that all deliveries are required. The delivery time cost of the shipment door-to-door, $mT_t$, indicates the delivery time cost of all the deliveries to the courier. According to the survey courier's distribution speed, the average time of delivery, and storage to the courier cabinet, in the following simulation experiment, the value of $v$ is 30 km/h, the value of $T_s$ is 10 min, and the value of $T_t$ is 5 min.

## 4. The Performance Analysis

### 4.1. Experimental Setup

In the experiment, the processing steps are as follows:

(1) Set up all the shipments that need to be delivered on the same day of the express center to establish a Cartesian coordinate system according to the latitude and longitude of the delivery address;

(2) Then, cluster according to the k-means algorithm, where the value of k is the number of couriers, and the deliveries in each class will be delivered by a courier.

(3) After clustering, plan the dispatch order of the express mail in each class according to the ant colony algorithm to find the shortest delivery distance.

(4) Calculate the delivery time cost according to the delivery distance of each class and the delivery method of each shipment, and then assign each courier class to 1 to k couriers according to the time cost.

(5) Calculate the standard deviation of the time price of all couriers as of the date based on the time cost and make a standard deviation line chart.

(6) Repeat Steps 1–5 and make the appropriate analysis.

### 4.2. Experimental Data Analysis

First of all, all the shipments that need to be dispatched that day will be clustered according to the delivery address.The first shipment's delivery address and the plane coordinates converted from latitude, longitude, and the number of shipments to be delivered for each address are shown in the Table 1:

**Table 1.** First day receipt information form.

| Recipient Address | Abs | Ord | DM | DN |
|---|---|---|---|---|
| Shanghai Open University | 12548 | 298408 | EC | 3 |
| Huangxing Park | 36555 | 299211 | DTH | 5 |
| Cultural Garden | 31669 | 294952 | EC | 3 |
| Jiangjiang Village | 36555 | 293535 | DTH | 2 |
| Tumen Community | 52869 | 300075 | DTH | 4 |
| Fengcheng Sancun | 28075 | 287855 | DTH | 3 |
| Changbai Sancun | 29066 | 299409 | DTH | 2 |
| Shanghai University of Technology | 51991 | 300890 | EC | 2 |
| Oriental Garden | 30648 | 287375 | EC | 4 |
| Yangpu Apartment | 35678 | 288023 | DTH | 4 |
| Yangpu Park | 40619 | 287066 | DTH | 2 |
| Yanji Middle School | 49117 | 289396 | EC | 4 |
| Fudan Software Park | 48416 | 283425 | EC | 3 |
| Democratic Second Village | 52584 | 289304 | DTH | 1 |
| Yangpu Auto Parts City | 57435 | 290877 | EC | 3 |
| Zhongxuan Liyuan | 58333 | 285138 | DTH | 4 |
| Changyang Chuanggu | 40349 | 278263 | DTH | 2 |
| Liaoyuan Second Village | 18718 | 275516 | DTH | 4 |
| Jiangpu Park | 30540 | 271871 | EC | 3 |
| Shuiyuefang | 48290 | 276099 | EC | 3 |
| Yinhe Court | 41930 | 273692 | EC | 2 |
| Yangpu Station | 54291 | 290048 | EC | 4 |
| Yinxianggang Community | 45523 | 278599 | DTH | 3 |
| Chang Xin Xin Yuan | 51201 | 279988 | DTH | 2 |
| East Liaoyang Middle School | 57633 | 280049 | EC | 3 |
| Wentong Building | 22203 | 269340 | EC | 6 |
| Phoenix Building | 20511 | 281037 | EC | 4 |
| Huayuan Haoting | 20515 | 283764 | DTH | 2 |
| Shuangyang Second Village | 35103 | 285369 | DTH | 5 |
| Home Inns | 45272 | 284536 | EC | 3 |
| Baiyangdian Football Stadium | 52782 | 288332 | EC | 2 |
| Oriental home | 21090 | 297033 | EC | 4 |
| Yuyuan | 23317 | 293423 | EC | 3 |
| Shanghai Business School | 24539 | 292158 | EC | 6 |
| Fengcheng Five Villages | 24359 | 287992 | DTH | 4 |
| Wanchang Building | 36939 | 289628 | DTH | 4 |
| Taihong Xinyuan | 41319 | 290646 | DTH | 1 |
| Yanji Nursing Home | 48039 | 292590 | EC | 5 |
| Neijiang Park | 51344 | 293115 | EC | 3 |
| Guangyuan Xincun | 54722 | 294010 | DTH | 4 |
| Donghai Medicine | 66436 | 297712 | EC | 5 |
| Express center | 38553 | 277870 | EC | 0 |

**Table 1.** *Cont.*

| Recipient Address | Abs | Ord | DM | DN |
|---|---|---|---|---|
| Meizhoufang | 36001 | 274413 | DTH | 3 |
| City business building | 47033 | 274567 | EC | 5 |
| Yangpu Central Hospital | 56662 | 278117 | EC | 4 |
| Bailian Riverside Shopping Center | 44302 | 271278 | EC | 5 |
| Yangpu District People's Court | 36828 | 269859 | EC | 6 |
| Huajing Hotel | 42613 | 276928 | EC | 2 |
| Siping Science Park | 12681 | 293236 | EC | 4 |
| Tongji New Village | 16346 | 289009 | DTH | 5 |
| Global Building | 38301 | 294409 | EC | 3 |
| National Science Building | 38660 | 286540 | EC | 1 |
| Environmental protection square | 42182 | 298852 | EC | 1 |
| Convenience market | 52710 | 291878 | EC | 3 |
| Sanfeng Building | 19221 | 278855 | EC | 2 |
| Wuhuan Building | 15053 | 273269 | EC | 3 |
| Yangpu Junior High School | 20874 | 273269 | EC | 4 |
| Shanghai Cigarette Factory | 24674 | 269072 | EC | 3 |
| Forest garden | 21233 | 266124 | EC | 2 |
| North American Square | 19616 | 265815 | DTH | 2 |
| Double Happiness Home | 20026 | 260089 | DTH | 1 |
| Yunmu Hotel | 27414 | 264411 | EC | 3 |
| Mingyuan Village Community | 28779 | 268470 | DTH | 2 |
| Huayuan Haoting | 20982 | 282096 | DTH | 1 |
| Tongji Green Park | 18143 | 280460 | DTH | 2 |
| City Concept Creative Park | 45955 | 281818 | EC | 2 |
| Vibrating | 55081 | 275121 | EC | 3 |
| Shun Chengli | 48470 | 269133 | EC | 2 |
| Donghuali | 45380 | 263546 | EC | 1 |
| Tianke International Building | 39092 | 266448 | EC | 3 |
| Shuyayuan Nursing Home | 40565 | 269442 | DTH | 2 |
| Miaojia cottage | 38050 | 273609 | EC | 2 |
| Yangpu District Children's Palace | 41032 | 274874 | EC | 1 |

In Table 1 and Figure 1, Abs is Abscissa, Ord is Ordinate, DM is delivery method, DN is delivery number. In the balanced k-means algorithm below, because the difference between the latitude and longitude of the receiving address is very small, we carried out a reasonable amplification and translation transformation on the data pre-processing, and then converted it into the logical distance calculated in the form of two-dimensional cartesian coordinates, so no specific distance unit is set for it in this paper. Because the latitude and longitude of the delivery addresses in this paper are not very different, the height difference between the delivery addresses is ignored. The specific method is to collect the longitude and latitude coordinates of the address first, convert the collected coordinates into two-dimensional cartesian coordinates, and then calculate the distance information according to the Euclidean distance calculation formula.

The coordinate information in the above table is converted from the latitude and longitude information of the packaged address of the package on the first day of the express center. The cluster analysis is performed by the k-means algorithm in order to obtain Figure 1.

As can be seen from Figure 1, all the express items that need to be delivered that day are clustered into five clusters. These clusters were allocated to five couriers for delivery. These five clusters are described in detail below.

As can be seen from the below figure, the receiving addresses in the first category are: Yiyuan, Changbai Sancun, Oriental Home, Shanghai Open University, Fengcheng Sancun-Shanghai Business School, Siping Science Park, Tongji New Village, and Fengcheng Five Villages.

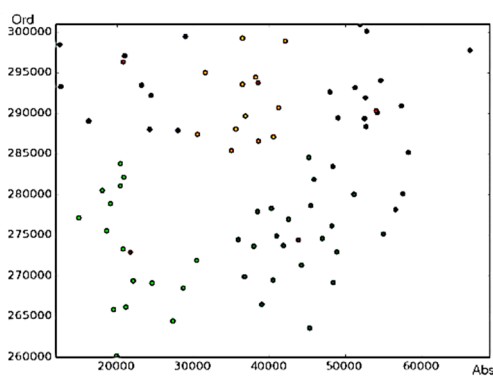

**Figure 1.** First day clustering.

The receiving addresses in the second category are: Cultural Garden, Global Building, Control River Five Village, Oriental Garden, Shuangyang Second Village, Express Center, Guoke Building, Yangpu Park, Wanchang Building, Yangpu Apartment ,Taihong Xinyuan, Environmental Protection Plaza and Huangxing Park.

The receiving addresses in the third category are: Yangpu Station, Democratic Second Village, Donghai Medicine, Yangpu Auto Parts City, Baiyangdian Football Stadium, Yangpu Central Hospital, Express Center, Zhongxuan Liyuan, Dongliaoyang Middle School, Fudan Software Park, Yanji Middle School, Yanji Nursing Home, Neijiang Park, Convenience Market, Guangyuan New Village, Tumen Community, and Shanghai University of Technology.

The receiving addresses in the fourth category are: Liaoyuan Ercun, Yangpu Junior High School, Sanfeng Building, Wuhuan Building, Wentong Building, Double Happiness Home, Yunmu Hotel, Mingyuan Village Community, Shanghai Cigarette Factory, Linyuan, North America Square, Jiangpu Park, Express Center, Huayuan Haoting, Huayuan Haoting, Fenghuang Building, and Tongji Green Park.

The receiving addresses in the fifth category includes: City Business Building, Shuiyuefang, Yinxianggang Community, Bailian Riverside Shopping Center, City Concept Creative Park, Rujia Hotel, Dong Huali, Tianke International Building, Shuyayuan Senior Citizen's Court, Changxin Xinyuan, Zhenshengli, Shunchengli, Yangpu District People's Court, Miaojia Shanzhai, Meizhoufang, Express Center, Changyang Chuanggu, Huajing Hotel, Yangpu District Children's Palace, and Yinhe Court.

In the distribution sequence diagram below, the numbers represent the distribution sequence, and the horizontal and vertical coordinates in the diagram correspond to the data in ordinate Table 1.

Using the ant colony algorithm to plan the dispatch order of the shipments in the first category.

It can be concluded from Figure 2 that the delivery order of the first batch of express mail is: Express center to Fengcheng Sancun to Shanghai Business School to Yuyuan to Changbai Sancun to Oriental Home to Shanghai Open University to Siping Science Park to Tongji New Village to Fengcheng Five Village.

Google Maps was used to get the delivery distance of the first shipment, 14.68 km. According to the delivery route of the first batch of express deliveries and the dispatch information in Table 1, the total delivery time of the first shipment is 4.48 h.

Using the ant colony algorithm to plan the dispatch order of the shipments in the second category:

It can be concluded from Figure 3 that the delivery order of the second batch of express mail is: Express Center to Guoke Building to Yangpu Park to Taihong Xinyuan to Environmental Protection Plaza to Huangxing Park to Global Building to Jiangjiang Village to Cultural Garden to Oriental Garden to Wanchang Building to Yangpu Apartment to Shuangyang Ercun.

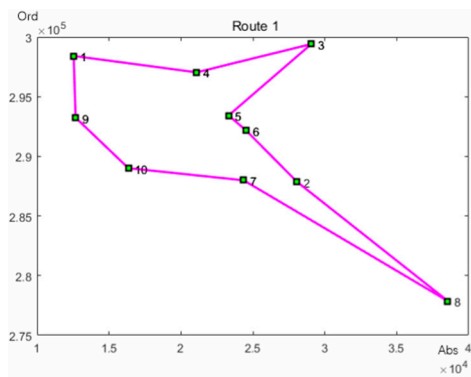

**Figure 2.** Delivery order of first courier.

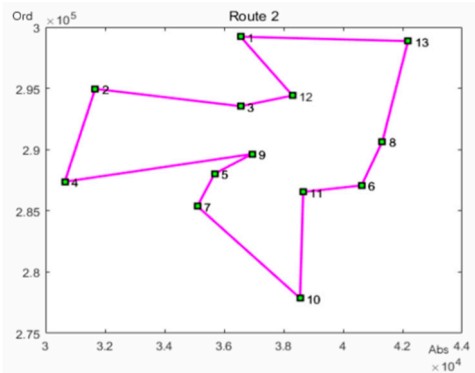

**Figure 3.** Delivery order of second courier.

With Google Maps, the delivery distance is 12.4 km for the second shipment. According to the delivery route of the second batch of express deliveries and the dispatch information in Table 1, the total delivery time of the second batch of express shipments is 5.24 h.

Using the ant colony algorithm to plan the dispatch order of the shipments in the third category:

It can be concluded from Figure 4 that the order of delivery of the third batch of express mail is: Express Center to Fudan Software Park to Yanji Middle School to Yanji Nursing Home to Neijiang Park to Convenience Market to Guangyuan New Village to Tumen Community to Shanghai University of Technology to Donghai Medicine to Yangpu Auto Parts City to Yangpu Station to Democracy 2 Village to Baiyangdian Football Stadium to Zhongxuan Liyuan to Dongliaoyang Middle School to Yangpu Central Hospital.

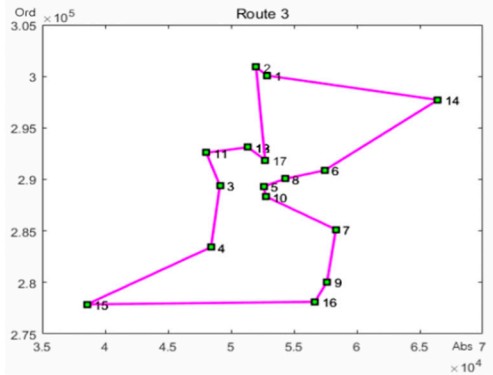

**Figure 4.** Delivery order of third courier.

With Google Maps, the delivery distance is 15.95 km for the third shipment. According to the delivery route of the third batch of express delivery and the dispatch information in Table 1, the total delivery time of the third batch of express shipments is 6.11 h.

Using the ant colony algorithm to plan the dispatch order of the shipments in the fourth category.

It can be concluded from Figure 5 that the delivery order of the fourth batch of express mail is: Express center to Huayuan Haoting to Huayuan Haoting to Phoenix Building to Tongji Green Park to Sanfeng Building to Wuhuan Building to Liaoyuan 2 Village to Yangpu Junior High School Wentong Building to Shanghai Cigarette Factory to Linyuan to North America Square to Double Happiness Home to Yunmu Hotel to Mingyuan Village Community to Jiangpu Park.

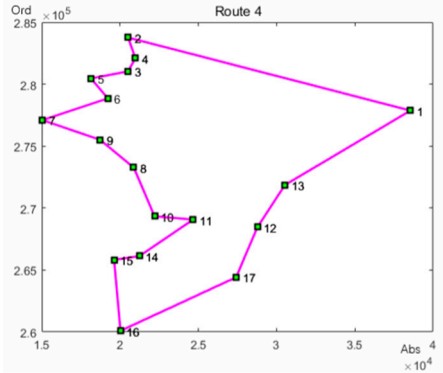

**Figure 5.** Delivery order of fourth courier.

With Google Maps, the delivery distance is 13.21 km for the fourth shipment. According to the delivery route of the four batches of express delivery and the dispatch information in Table 1, the total delivery time of the four batches of express shipments is 5.27 h.

Using the ant colony algorithm to plan the dispatch order of the shipments in the fifth category.

It can be concluded from Figure 6 that the delivery order of the fifth batch of express mail is: Express center to Changyang Chuanggu to Huajing Hotel to Yangpu District Children's Palace to Yinheyuan Bailian Riverside Shopping Center to City Business Building to Shuiyuefang to Yinxianggang Community Go to the City Concept Creative Park to the Home Inn to Changxin Xinyuan to Zhenshengli to Shunchengli to Donghuali to Tianke International Building to Shuyayuan Retirement House to Yangpu District People's Court to Miaojia Village to Meizhou Square.

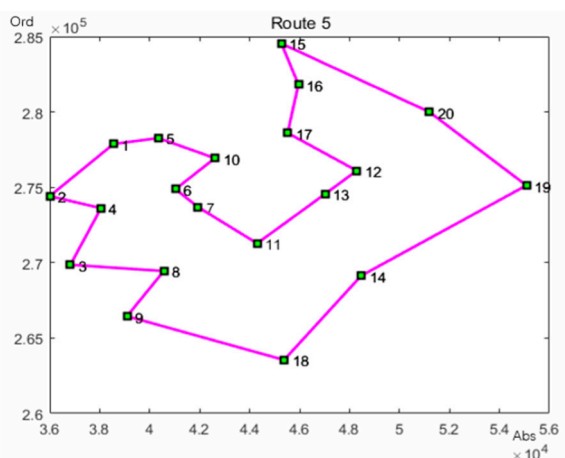

**Figure 6.** Delivery order of fifth courier.

With Google Maps, the delivery distance is 12.93 km for the fifth shipment. According to the delivery route of the five batch of express delivery and the dispatch information in Table 1, the total delivery time of the five batch of express shipments is 5.76 h.

The article has calculated the total cost of delivery time for the five batches of couriers obtained on the first day of clustering. The five batches of express mail will be distributed, from the big to the small, to couriers No. 1 to No. 5 according to the delivery time. On the first day, the cumulative time for the courier No. 1 to the courier No. 5 is shown in Table 2:

**Table 2.** Cumulative time.

| Courier 1 | Courier 2 | Courier 3 | Courier 4 | Courier 5 |
|-----------|-----------|-----------|-----------|-----------|
| 6.11 h    | 5.76 h    | 5.27 h    | 5.24 h    | 4.48 h    |

Calculated by standard deviation:

$$\sigma r = \sqrt{\frac{1}{N} \sum_{i=1}^{N} (x_i - r)^2},$$

(14)

The standard deviation of the cumulative time of the five couriers on the first day was: 0.5511.

The above steps were repeated to calculate the total cost of the delivery time of the five batches of express delivery from the second day to the fifteenth day. When assigning a dispatch task according to the dispatch time of the class, the courier class of the day is allocated in descending order of the delivery time cost to the courier who has accumulated the delivery time. Then, the total cost of the delivery time accumulated by the five couriers from the first day to the fifteenth day is calculated, and the standard deviation of the accumulated time of the five couriers is calculated too.

In the below table, Table 3 shows the accumulated delivery time cost of the first courier in the 15 days. Table 4 shows the cumulative delivery time cost of the second Courier in the 15 days. Table 5 shows the cumulative delivery time cost of the third Courier during the 15 days. Table 6 shows the cumulative delivery time cost of the fourth Courier in the 15 days. Table 7 shows the cumulative delivery time cost of the sixth Courier in the 15 days. Table 8 shows the standard deviation of the five couriers' cumulative delivery time cost in each of the 15 days.

**Table 3.** Courier 1 cumulative time cost.

| Days | Courier Number | Cumulative Time Cost (h) |
|------|----------------|--------------------------|
| 1    | 1              | 6.11                     |
| 2    | 1              | 11.61                    |
| 3    | 1              | 15.61                    |
| 4    | 1              | 23.61                    |
| 5    | 1              | 28.61                    |
| 6    | 1              | 35.81                    |
| 7    | 1              | 42.31                    |
| 8    | 1              | 48.61                    |
| 9    | 1              | 54.91                    |
| 10   | 1              | 60.81                    |
| 11   | 1              | 67.51                    |
| 12   | 1              | 74.41                    |
| 13   | 1              | 81.81                    |
| 14   | 1              | 89.69                    |
| 15   | 1              | 96.59                    |

**Table 4.** Courier 2 cumulative time cost.

| Days | Courier Number | Cumulative Time Cost (h) |
|------|----------------|--------------------------|
| 1    | 2              | 5.76                     |
| 2    | 2              | 11.46                    |
| 3    | 2              | 15.96                    |
| 4    | 2              | 23.46                    |
| 5    | 2              | 28.96                    |
| 6    | 2              | 35.76                    |
| 7    | 2              | 42.76                    |
| 8    | 2              | 48.26                    |
| 9    | 2              | 54.96                    |
| 10   | 2              | 60.66                    |
| 11   | 2              | 67.56                    |
| 12   | 2              | 74.36                    |
| 13   | 2              | 82.26                    |
| 14   | 2              | 89.46                    |
| 15   | 2              | 97.36                    |

**Table 5.** Courier 3 cumulative time cost.

| Days | Courier Number | Cumulative Time Cost (h) |
|------|----------------|--------------------------|
| 1    | 3              | 5.27                     |
| 2    | 3              | 11.07                    |
| 3    | 3              | 16.57                    |
| 4    | 3              | 22.57                    |
| 5    | 3              | 29.07                    |
| 6    | 3              | 35.57                    |
| 7    | 3              | 43.17                    |
| 8    | 3              | 47.79                    |
| 9    | 3              | 54.89                    |
| 10   | 3              | 61.19                    |
| 11   | 3              | 67.49                    |
| 12   | 3              | 74.69                    |
| 13   | 3              | 81.49                    |
| 14   | 3              | 89.62                    |
| 15   | 3              | 97.16                    |

**Table 6.** Courier 4 cumulative time cost.

| Days | Courier Number | Cumulative Time Cost (h) |
|------|----------------|--------------------------|
| 1    | 4              | 5.24                     |
| 2    | 4              | 11.44                    |
| 3    | 4              | 16.44                    |
| 4    | 4              | 23.44                    |
| 5    | 4              | 29.44                    |
| 6    | 4              | 35.44                    |
| 7    | 4              | 42.64                    |
| 8    | 4              | 48.74                    |
| 9    | 4              | 54.54                    |
| 10   | 4              | 61.24                    |
| 11   | 4              | 67.16                    |
| 12   | 4              | 75.06                    |
| 13   | 4              | 81.56                    |
| 14   | 4              | 89.66                    |
| 15   | 4              | 96.94                    |

**Table 7.** Courier 5 cumulative time cost.

| Days | Courier Number | Cumulative Time Cost (h) |
|---|---|---|
| 1 | 5 | 4.48 |
| 2 | 5 | 10.88 |
| 3 | 5 | 16.88 |
| 4 | 5 | 21.88 |
| 5 | 5 | 28.88 |
| 6 | 5 | 35.88 |
| 7 | 5 | 41.88 |
| 8 | 5 | 48.98 |
| 9 | 5 | 54.18 |
| 10 | 5 | 61.38 |
| 11 | 5 | 67.18 |
| 12 | 5 | 74.68 |
| 13 | 5 | 81.88 |
| 14 | 5 | 89.38 |
| 15 | 5 | 97.38 |

**Table 8.** Deliverer's cumulative delivery time standard deviation.

| Days | Standard Deviation |
|---|---|
| 1 | 0.5511 |
| 2 | 0.2721 |
| 3 | 0.4517 |
| 4 | 0.6657 |
| 5 | 0.2707 |
| 6 | 0.1627 |
| 7 | 0.4145 |
| 8 | 0.4343 |
| 9 | 0.2979 |
| 10 | 0.2735 |
| 11 | 0.1731 |
| 12 | 0.2497 |
| 13 | 0.2727 |
| 14 | 0.1207 |
| 15 | 0.2950 |

*4.3. Discussion of Experimental Results*

From the above data, we can get the daily delivery time histogram and the cumulative delivery time standard deviation line chart of each courier.

Since the courier center does not have a good plan for the task assignment of the dispatcher, the courier's delivery order and route are more uncertain when the mail is delivered. During the data collection, because the couriers chose the delivery route randomly, we have collected the address information and specific income of the couriers, and their actual delivery time cost is temporarily unknown.

According to the task assignment method of this article, the delivery time cost of each courier is obtained, and then the salary will be set proportionally according to the delivery time cost. Next, we compare the changes in the actual income of the dispatchers collected before the experiment with the corresponding indicators of the test delivery time, and then draw a reliable conclusion.

The income of the five couriers in the express delivery center within the same 15 days was collected before the experiment, and a line chart of the difference between the income and income standard was drawn. Then, the income graph and time cost graph was compared and analyzed.

Figure 7 illustrates the change of the accumulated income of five couriers from the express center in the past 15 days. Figure 8 shows the change diagram of the cumulative delivery time cost of five couriers in these 15 days calculated through experiments. Figure 9 shows the line graph obtained after

normalizing the standard deviation of the courier's accumulated daily income. Figure 10 is a line graph normalized by the standard deviation of the courier's cumulative delivery time cost per day.

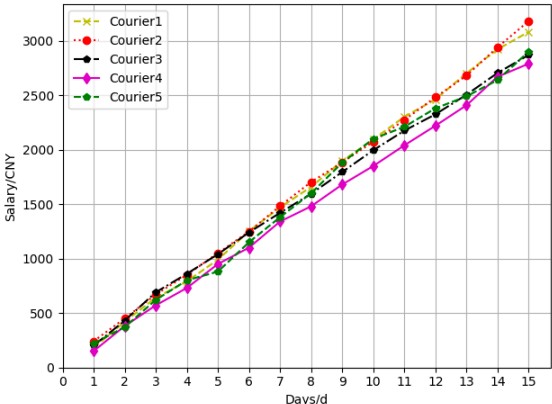

**Figure 7.** Cumulative Salary chart.

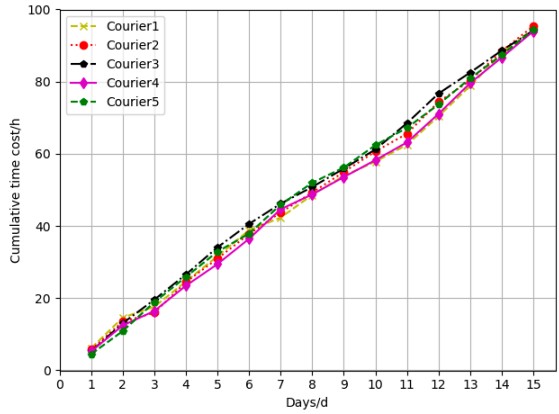

**Figure 8.** Cumulative time cost diagram.

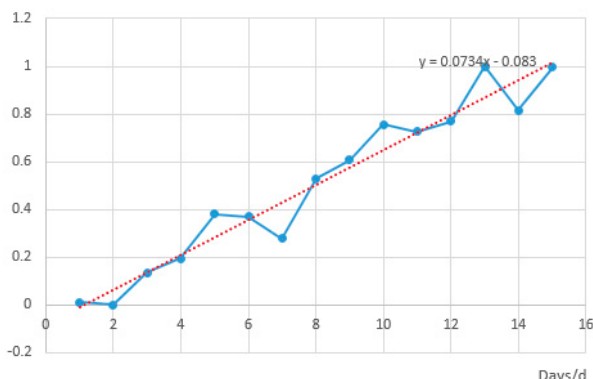

**Figure 9.** Original Standard deviation chart.

As can be seen from Figure 7, the cumulative income between the five couriers in the 15 days from the express center is uneven, and the cumulative income gap between couriers is relatively large. It can be seen from Figure 8 that the cumulative difference of delivery time cost between five couriers in the 15 days calculated through the experiment is very small, and the income of the couriers is well distributed using the proportion of the delivery time cost. It can be seen from Figure 9 that there is a big difference in the income of the deliverymen according to the statistics of the express center, and the trend line is $y = 0.0734x - 0.083$. The standard deviation is increasingly larger, and there is no stable trend. As can be seen from Figure 10, the trend line of the standard deviation of the cumulative

delivery time of dispatchers is y = −0.202ln(x) + 0.7543, and the standard deviation gradually decreases and tends to stabilize. The below information can indicate that the task assignment scheme in this paper is better than the previous one, and the task assignment scheme is stable and effective [7,52].

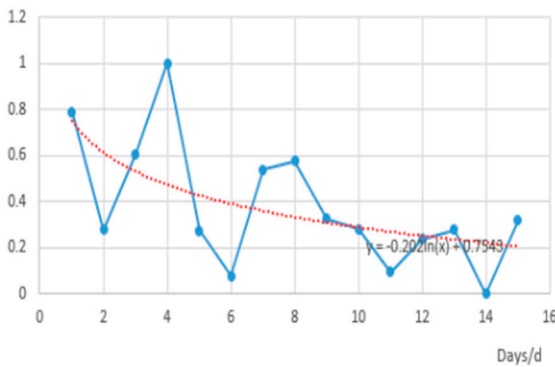

**Figure 10.** Standard deviation chart.

*4.4. Industrial Applications*

According to the experimental results of this paper, it can be seen that the task assignment scheme for the modern logistics terminal is effective and feasible. Therefore, this paper, combined with the real situation of logistics delivery, designed a specific industrial design scheme. The scheme is shown in Figure 11.

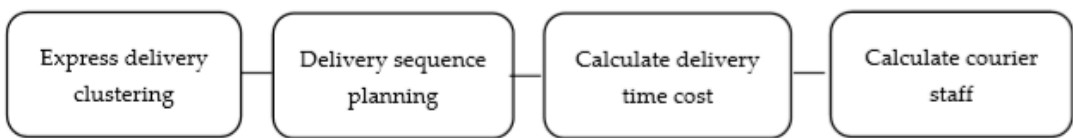

**Figure 11.** Industrial applications.

When the express center receives all the express mail that needs to be delivered on that day, it will cluster the express mail according to the longitude and latitude information of the delivery addresses and the balanced k-means algorithm, according to the number of express deliveries and couriers.

After clustering all express items that need to be delivered on that day, the ant colony algorithm is used to regard it as a TSP. The delivery sequence planning is carried out for express items in these classes, respectively, and the optimal delivery sequence of express items in each class is worked out to minimize the delivery distance.

The total delivery time cost of each delivery class is calculated according to the delivery distance in each delivery class and the delivery mode between different addresses and other factors affecting the delivery time. The delivery class is assigned to each courier according to the level of delivery time cost.

Complete the assignment task one day through the above steps, repeat the above steps the next day, and then assign the delivery time cost from low to high to the deliveryman whose cumulative delivery time cost is from high to low. The cycle is one month.

According to the delivery time cost of each type of delivery, combined with the local income status, the delivery staff's salary is allocated in equal proportion, so as to make the wage gap between the delivery staff very small.

## 5. Conclusions and Future Works

According to the simulation experiment in the fourth section, it can be seen that, in the 15-day express delivery of the simulation experiment, the difference of the total cost of delivery time between each courier is small, and the difference between the cumulative delivery time of each courier is small. The trend line of the standard deviation scatter plot of cumulative delivery time of the five couriers is y

= −0.202ln(x) + 0.7543. This trend line indicates that the standard deviation is getting smaller and tends to stabilize, which means that the task assignment method designed by the express center for the courier dispatcher has a good effect, which can solve the unbalanced distribution of the courier dispatch task problem, such that the express center can more rationally allocate the workload and income of each courier, which can not only improve the delivery efficiency of the courier center, but also promote the development of the courier company. Further, it can effectively make the delivery workload between the distributors roughly the same and improve the delivery efficiency of the individual couriers, save energy, and promote sustainable development.

In future works, we will continue to optimize the clustering algorithm in order to achieve better clustering results, improve the ant colony algorithm, and make it more efficient to plan the delivery sequence. For the time-cost calculation model, we will take into account more influencing factors and make the algorithm closer to the actual situation. For example, weather factors, road congestion, different weather conditions, and road traffic conditions will affect the delivery efficiency of couriers. The paper only records the task assignment of five couriers within 15 days. In the future, we will consider the task assignment of more couriers for a longer period of time, so as to make the experimental results obtained in this paper more realistic and convincing.

**Author Contributions:** Conceptualization, J.W. and C.W.; methodology, J.W. and C.W.; software, J.W., Q.W. and N.N.X.; validation, J.W., Y.W. and X.L. formal analysis, C.W., Y.W. and N.N.X.; investigation, J.W. and Y.W.; resources, C.W. and N.N.X.; data curation, J.W. and Q.W.; writing—original draft preparation, J.W.; writing—review and editing, C.W., Y.W. and N.N.X.; visualization, J.W. and Q.W.; supervision, C.W. and N.N.X.; project administration, C.W. and N.N.X.

**Funding:** This research was funded by the National Key Research and Development Program of China (No. 2018YFC0810204, 2018YFB17026), National Natural Science Foundation of China (No. 61872242), Shanghai Science and Technology Innovation Action Plan Project (17511107203) and Shanghai key lab of modern optical system.

**Acknowledgments:** The authors would like to appreciate all anonymous reviewers for their insightful comments and constructive suggestions to polish this paper in high quality.

**Conflicts of Interest:** All authors declare no conflicts of interest in this paper.

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
