# Peer review of "Design and Analysis of the Task Distribution Scheme of Express Center at the End of Modern Logistics"

_electronics, doi:10.3390/electronics8101141_

Round 1

Reviewer 1 Report

   The study presents an algorithm for calculating the delivery time cost of express delivery and uses the ant colony algorithm to plan the dispatch order of the shipments. The results show that the task assignment scheme for  modern logistics terminal is effective and feasible.

Although the limited factors considered in the model, the authors have the intention to continue to optimize the algorithm, improving the ant colony algorithm and make it more efficiente, taking into account other influencing factors. The article present a very recent bibliography The model can be applied and suggests that the method has a good effect on the dispatching task assignment of the express center Some corrections need to be done: Line 294: missing an empty line before title Line 299: missing an empty line after table Line 303: Change “FIGURE” by “Figure” Line 401: missing an empty line after figure Line 422: missing an empty line before title

Author Response

    Thank you very much for your valuable advice. I will continue to optimize the algorithm in future work to make it more efficient. I have revised the corresponding part of the paper according to your opinion. Line 294: add an empty line before title. Line 299: add an empty line before title. Line 303: Change “FIGURE” by “Figure”. Line 401: add an empty line after figure Line. Line 422: add an empty line after figure Line.

Reviewer 2 Report

Abstract: Assigning scientific task to each courier dispatcher is the research question addressed in this paper, and not the problem.

The literature review section currently lists the contribution of previous authors one by one. There should be a paragraph at the end of this section explaining the research gaps identified from the review and how this paper addresses these research gaps.

In the Balanced K-means algorithm, what is the Euclidean distance between two delivery addresses? Is it the actual geographical distance between two addresses or the addresses are first converted into a two-dimensional cartesian form and then a straight-line distance is assumed? Looking at section 4.1 point #1, it looks like the authors have used cartesian distances. Which projection scheme was used to convert latitude longitude into X and Y coordinates? Please describe in the paper.

The authors can avoid explaining K-means and Balanced K-means algorithm in detail in this paper as these are already very established algorithms. I would suggest only explain the algorithm with respect to any deviations assumed from the existing algorithm. Similar comment applies to explaining of the Ant Colony algorithm in section 3.2.

There is no justification provided by the authors for using ant colony optimization for TSP? Authors have only mentioned about some of the previous papers who have used ant colony optimization for vehicle routing problems.

Paragraphs below figure 1 is adding no value to the manuscript as the “Abs” and “Ord” values are described in the table and it is difficult to identify each location in the figure using those values.

Does Figure 1 shows distinctive clusters? How many clusters were finally identified?

Similarly, for all the routing figures (Figure 2 through 6), what does the number on the nodes represent? How the explanation of different locations in the route correspond to these number on the nodes. To be concise, explanation of these all figures can be avoided by describing each route in detail.

How the results of time cost from this methodology compare to the actual time cost based on real time data? This should be clearly explained in the results section and should be tied to the research gaps/existing contributions in the field.

There are some typos in the manuscript. For example, see line 304 (“figureure”).

Round 2

Reviewer 2 Report

Thanks for making the suggested changes